# Dual T: Reducing Estimation Error for Transition Matrix in Label-noise Learning

**Yu Yao**[1]    **Tongliang Liu**[1][†]    **Bo Han**[2]
**Mingming Gong**[3]    **Jiankang Deng**[4]    **Gang Niu**[5]    **Masashi Sugiyama**[5,6]

[1]University of Sydney; [2]Hong Kong Baptist University; [3]University of Melbourne;
[4]Imperial College London; [5]RIKEN; [6]University of Tokyo

## Abstract

The *transition matrix*, denoting the transition relationship from clean labels to noisy labels, is essential to build *statistically consistent* classifiers in label-noise learning. Existing methods for estimating the transition matrix rely heavily on estimating the noisy class posterior. However, the estimation error for *noisy class posterior* could be large due to the randomness of label noise, which would lead the transition matrix to be poorly estimated. Therefore, in this paper, we aim to solve this problem by exploiting the divide-and-conquer paradigm. Specifically, we introduce an *intermediate class* to avoid directly estimating the noisy class posterior. By this intermediate class, the original transition matrix can then be factorized into the product of two easy-to-estimate transition matrices. We term the proposed method the *dual-T estimator*. Both theoretical analyses and empirical results illustrate the effectiveness of the dual-$T$ estimator for estimating transition matrices, leading to better classification performances.

## 1 Introduction

Deep learning algorithms rely heavily on large annotated training samples [6]. However, it is often expensive and sometimes infeasible to annotate large datasets accurately [11]. Therefore, cheap datasets with label noise have been widely employed to train deep learning models [40]. Recent results show that label noise significantly degenerates the performance of deep learning models, as deep neural networks can easily memorize and eventually fit label noise [46, 1].

Existing methods for learning with noisy labels can be divided into two categories: algorithms with *statistically inconsistent* or *consistent* classifiers. Methods in the first category usually employ heuristics to reduce the side-effects of label noise, such as extracting reliable examples [12, 42, 43, 11, 23, 29, 13], correcting labels [22, 14, 32, 28], and adding regularization [10, 9, 35, 34, 18, 17, 36]. Although those methods empirically work well, the classifiers learned from noisy data are not guaranteed to be statistically consistent. To address this limitation, algorithms in the second category have been proposed. They aim to design *classifier-consistent* algorithms [44, 48, 14, 19, 25, 30, 24, 8, 26, 33, 45, 20, 41, 38], where classifiers learned by exploiting noisy data will asymptotically converge to the optimal classifiers defined on the clean domain. Intuitively, when facing large-scale noisy data, models trained via classifier-consistent algorithms will approximate to the optimal models trained with clean data.

The *transition matrix* $T(\boldsymbol{x})$ plays an essential role in designing *statistically consistent* algorithms, where $T_{ij}(\boldsymbol{x}) = P(\bar{Y} = j | Y = i, X = \boldsymbol{x})$ and we set $P(A)$ as the probability of the event $A$, $X$ as the random variable of instances/features, $\bar{Y}$ as the variable for the noisy label, and $Y$ as the

---

[†]Correspondence to Tongliang Liu (tongliang.liu@sydney.edu.au).

variable for the clean label. The basic idea is that the clean class posterior can be inferred by using the transition matrix and noisy class posterior (which can be estimated by using noisy data). In general, the transition matrix $T(\boldsymbol{x})$ is unidentifiable and thus hard to learn [4, 37]. Current state-of-the-art methods [11, 10, 26, 25, 24] assume that the transition matrix is *class-dependent* and *instance-independent*, i.e., $P(\bar{Y} = j|Y = i, X = \boldsymbol{x}) = P(\bar{Y} = j|Y = i)$. Given *anchor points* (the data points that belong to a specific class almost surely), the class-dependent and instance-independent transition matrix is identifiable [19, 30], and it could be estimated by exploiting the noisy class posterior of anchor points [19, 26, 45] (more details can be found in Section 2). In this paper, we will focus on learning the class-dependent and instance-independent transition matrix which can be used to improve the classification accuracy of the current methods if the matrix is learned more accurately.

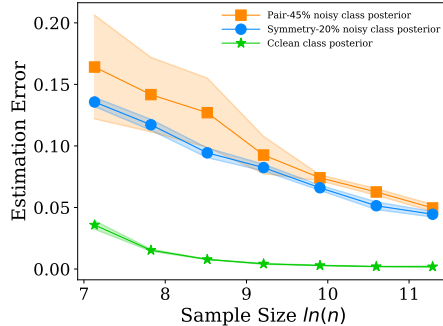

The estimation error for the noisy class posterior is usually much larger than that of the clean class posterior, especially when the sample size is limited. An illustrative example is in Fig. 1. The rationale is that label noise are randomly generated according to a class-dependent transition matrix. Specifically, to learn the noisy class posterior, we need to fit the mapping from instances to (latent) clean labels, as well as the mapping from clean labels to noisy labels. Since the latter mapping is much more random and independent of instances, the learned mapping that fits label noise is prone to overfitting and thus will lead to a large estimation error for the noisy class posterior [46]. The error will also lead to a large estimation error for the transition matrix. As estimating the transition matrix is a bottleneck for designing consistent algorithms, the large estimation error will significantly degenerate the classification performance [37].

Figure 1: Estimation errors for clean class posteriors and noisy class posteriors on synthetic data. The estimation errors are calculated as the average absolute value between the ground-truth and estimated class posteriors on $1,000$ randomly sampled test data points. The other details are the same as those of the synthetic experiments in Section 4.

Motivated by this phenomenon, in this paper, to reduce the estimation error of the transition matrix, we propose the *dual transition estimator* (*dual-T estimator*) to effectively estimate transition matrices. In a high level, by properly introducing an intermediate class, the dual-$T$ estimator avoids directly estimating the noisy class posterior, via factorizing the original transition matrix into two new transition matrices, which we denote as $T^{\clubsuit}$ and $T^{\spadesuit}$. $T^{\clubsuit}$ represents the transition from the clean labels to the intermediate class labels and $T^{\spadesuit}$ the transition from the clean and intermediate class labels to the noisy labels. Note that although we are going to estimate two transition matrices rather than one, we are not transforming the original problem to a harder one. In philosophy, our idea belongs to the divide and conquer paradigm, i.e., decomposing a hard problem into simple sub-problems and composing the solutions of the sub-problems to solve the original problem. The two new transition matrices are easier to estimate than the original transition matrix, because we will show that (1) there is no estimation error for the transition matrix $T^{\clubsuit}$, (2) the estimation error for the transition matrix $T^{\spadesuit}$ relies on predicting noisy class labels, which is much easier than learning a class posterior, as predicting labels require much less information than estimating noisy posteriors[1], and (3) the estimators for the two new transition matrices are easy to implement in practice. We will also theoretically prove that the two new transition matrices are easier to predict than the original transition matrix. Empirical results on several datasets and label-noise settings consistently justify the effectiveness of the dual-$T$ estimator on reducing the estimation error of transition matrices and boosting the classification performance.

The rest of the paper is organized as follows. In Section 2, we review the current transition matrix estimator that exploits anchor points. In Section 3, we introduce our method and analyze how it reduces the estimation error. Experimental results on both synthetic and real-world datasets are provided in Section 4. Finally, we conclude the paper in Section 5.

## 2 Estimating Transition Matrix for Label-noise Learning

**Problem setup.** Let $D$ be the distribution of a pair of random variables $(X, Y) \in \mathcal{X} \times \{1, \ldots, C\}$, where $X$ denotes the variable of instances, $Y$ the variable of labels, $\mathcal{X}$ the feature space, $\{1, \ldots, C\}$ the label space, and $C$ the size of classes. In many real-world classification problems, examples independently drawn from $D$ are unavailable. Before being observed, their clean labels are randomly flipped into noisy labels because of, e.g., contamination [31]. Let $\bar{D}$ be the distribution of the noisy pair $(X, \bar{Y})$, where $\bar{Y}$ denotes the variable of noisy labels. In label-noise learning, we only have a sample set $\bar{S} = \{(\boldsymbol{x}_i, \bar{y}_i)\}_{i=1}^n$ independently drawn from $\bar{D}$. The aim is to learn a robust classifier from the noisy sample $\bar{S}$ that can assign clean labels for test instances.

**Transition matrix.** To build statistically consistent classifiers, which will converge to the optimal classifiers defined by using clean data, we need to introduce the concept of transition matrix $T(\boldsymbol{x}) \in \mathbb{R}^{C \times C}$ [24, 19, 28]. Specifically, the $ij$-th entry of the transition matrix, i.e., $T_{ij}(\boldsymbol{x}) = P(\bar{Y} = j | Y = i, X = \boldsymbol{x})$, represents the probability that the instance $\boldsymbol{x}$ with the clean label $Y = i$ will have a noisy label $\bar{Y} = j$. The transition matrix has been widely studied to build statistically consistent classifiers, because the clean class posterior $P(\boldsymbol{Y} | \boldsymbol{x}) = [P(Y = 1 | X = \boldsymbol{x}), \ldots, P(Y = C | X = \boldsymbol{x})]^\top$ can be inferred by using the transition matrix and the noisy class posterior $P(\bar{\boldsymbol{Y}} | \boldsymbol{x}) = [P(\bar{Y} = 1 | X = \boldsymbol{x}), \ldots, P(\bar{Y} = C | X = \boldsymbol{x})]^\top$, i.e., we have $P(\bar{\boldsymbol{Y}} | \boldsymbol{x}) = T(\boldsymbol{x}) P(\boldsymbol{Y} | \boldsymbol{x})$. Specifically, the transition matrix has been used to modify loss functions to build risk-consistent estimators, e.g., [8, 26, 45, 37], and has been used to correct hypotheses to build classifier-consistent algorithms, e.g., [24, 30, 26]. Moreover, the state-of-the-art statically inconsistent algorithms [13, 11] also use diagonal entries of the transition matrix to help select reliable examples used for training.

As the noisy class posterior can be estimated by exploiting the noisy training data, the key step remains how to effectively estimate the transition matrix. Given only noisy data, the transition matrix is unidentifiable without any knowledge on the clean label [37]. Specifically, the transition matrix can be decomposed to product of two new transition matrices, i.e., $T(\boldsymbol{x}) = T'(\boldsymbol{x}) A(\boldsymbol{x})$, and a different clean class posterior can be obtained by composing $P(\boldsymbol{Y} | \boldsymbol{x})$ with $A(\boldsymbol{x})$, i.e., $P'(\boldsymbol{Y} | \boldsymbol{x}) = A(\boldsymbol{x}) P(\boldsymbol{Y} | \boldsymbol{x})$. Therefore, $P(\bar{\boldsymbol{Y}} | \boldsymbol{x}) = T(\boldsymbol{x}) P(\boldsymbol{Y} | \boldsymbol{x}) = T'(\boldsymbol{x}) P'(\boldsymbol{Y} | \boldsymbol{x})$ are both valid decompositions. The current state-of-the-art methods [11, 10, 26, 25, 24] then studied a special case by assuming that the transition matrix is *class-dependent* and *instance-independent*, i.e., $T(\boldsymbol{x}) = T$. Note that there are specific settings [7, 21, 2] where noise is independent of instances. A series of assumptions [19, 30, 27] were further proposed to identify or efficiently estimate the transition matrix by only exploiting noisy data. In this paper, we focus on estimating the class-dependent and instance-independent transition matrix which is focused by vast majority of current state-of-the-art label-noise learning algorithms [11, 10, 26, 25, 24, 13, 11]. The estimated matrix by using our method then can be seamlessly embedded into these algorithms, and the classification accuracy of the algorithms can be improved, if the transition matrix is estimated more accurate.

**Transition matrix estimation.** The anchor point assumption [19, 30, 37] is a widely adopted assumption to estimate the transition matrix. Anchor points are defined in the clean data domain. Formally, an instance $\boldsymbol{x}^i \in \mathcal{X}$ is an anchor point of the $i$-th clean class if $P(Y = i | \boldsymbol{x}^i) = 1$ [19, 37]. Suppose we can assess to the the noisy class posterior and anchor points, the transition matrix can be obtained via $P(\bar{Y} = j | \boldsymbol{x}^i) = \sum_{k=1}^C P(\bar{Y} = j | Y = k, \boldsymbol{x}^i) P(Y = k | \boldsymbol{x}^i) = P(\bar{Y} = j | Y = i, \boldsymbol{x}) = T_{ij}$, where the second equation holds because $P(Y = k | \boldsymbol{x}^i) = 1$ when $k = i$ and $P(Y = k | \boldsymbol{x}^i) = 0$ otherwise. The last equation holds because the transition matrix is independent of the instance. According to the Equation, to estimate the transition matrix, we need to find anchor points and estimate the noisy class posterior, then the transition matrix can be estimated as follows,

$$\hat{P}(\bar{Y} = j | \boldsymbol{x}^i) = \sum_{k=1}^C \hat{P}(\bar{Y} = j | Y = k, \boldsymbol{x}^i) P(Y = k | \boldsymbol{x}^i) = \hat{P}(\bar{Y} = j | Y = i, \boldsymbol{x}) = \hat{P}_{ij}. \quad (1)$$

This estimation method has been widely used [19, 26, 37] in label-noise learning and we term it the *transition estimator* ($T$ estimator).

Note that some methods assume anchor points have already been given [45]. However, this assumption could be strong for applications, where anchor points are hard to identify. It has been proven that anchor points can be learned from noisy data [19], i.e., $\boldsymbol{x}^i = \arg\max_{\boldsymbol{x}} P(\bar{Y} = i | \boldsymbol{x})$, which only holds for binary classification. The same estimator has also been employed for multi-class classification

[26]. It empirically performs well but lacks theoretical guarantee. How to identify anchor points in the multi-class classification problem with theoretical guarantee remains an unsolved problem.

Eq. (1) and the above discussions on learning anchor points show that the $T$ estimator relies heavily on the estimation of the noisy class posterior. Unfortunately, due to the randomness of label noise, the estimation error of the noisy class posterior is usually large. As the example illustrated in Fig. 1, with the same number of training examples, the estimation error of the noisy class posterior is significantly larger than that of the clean class posterior. This motivates us to seek for an alternative estimator that avoids directly using the estimated noisy class posterior to approximate the transition matrix.

## 3   Reducing Estimation Error for Transition Matrix

To avoid directly using the estimated noisy class posterior to approximate the transition matrix, we propose a new estimator in this section.

### 3.1   dual-$T$ estimator

By introducing an intermediate class, the transition matrix $T$ can be factorized in the following way:

$$
\begin{aligned}
T_{ij} = P(\bar{Y} = j | Y = i) &= \sum_{l \in \{1, \ldots, C\}} P(\bar{Y} = j | Y' = l, Y = i) P(Y' = l | Y = i) \\
&\triangleq \sum_{l \in \{1, \ldots, C\}} T_{lj}^{\spadesuit}(Y = i) T_{il}^{\clubsuit} ,
\end{aligned}
\tag{2}
$$

where $Y'$ represent the random variable for the introduced intermediate class, $T_{lj}^{\spadesuit}(Y = i) = P(\bar{Y} = j | Y' = l, Y = i)$, and $T_{il}^{\clubsuit} = P(Y' = l | Y = i)$. Note that $T^{\spadesuit}$ and $T^{\clubsuit}$ are two transition matrices representing the transition from the clean and intermediate class labels to the noisy class labels and transition from the clean labels to the intermediate class labels, respectively.

By looking at Eq. (2), it seems we have changed an easy problem into a hard one. However, this is totally not true. Actually, we break down a problem into simple sub-problems. Combining the solutions to the sub-problems gives a solution to the original problem. Thus, in philosophy, our idea belongs to the divide and conquer paradigm. In the rest of this subsection, we will explain why it is easy to estimate the transition matrices $T^{\spadesuit}$ and $T^{\clubsuit}$. Moreover, in the next subsection, we will theoretically compare the estimation error of the dual-$T$ estimator with that of the $T$ estimator.

It can be found that $T_{ij}^{\clubsuit} = P(Y' = j | Y = i)$ has a similar form to $T_{ij} = P(\bar{Y} = j | Y = i)$. We can employ the same method that is developed for $T$, i.e., the $T$ estimator, to estimate $T^{\clubsuit}$. However, there seems to have two challenges: (1) it looks as if difficult to access $P(Y'|\boldsymbol{x})$; (2) we may also have an error for estimating $P(Y'|\boldsymbol{x})$. Fortunately, these two challenges can be well addressed by properly introducing the intermediate class. Specifically, we design the intermediate class $Y'$ in such a way that $P(Y'|\boldsymbol{x}) \triangleq \hat{P}(\bar{Y}|\boldsymbol{x})$, where $\hat{P}(\bar{Y}|\boldsymbol{x})$ represents an estimated noisy class posterior. Note that $\hat{P}(\bar{Y}|\boldsymbol{x})$ can be obtained by exploiting the noisy data at hand. As we have discussed, due to the randomness of label noise, estimating $T$ directly will have a large estimation error especially when the noisy training sample size is limited. However, as we have access to $P(Y'|\boldsymbol{x})$ directly, according to Eq. (1), the estimation error for $T^{\clubsuit}$ is zero if anchor points are given[2].

Although the transition matrix $T^{\spadesuit}$ contains three variables, i.e., the clean class, intermediate class, and noisy class, we have class labels available for two of them, i.e., the intermediate class and noisy class. Note that the intermediate class labels can be assigned by using $P(Y'|\boldsymbol{x})$. Usually, the clean class labels are not available. This motivates us to find a way to eliminate the dependence on clean class for $T^{\spadesuit}$. From an information-theoretic point of view [5], if the clean class $Y$ is less informative for the noisy class $\bar{Y}$ than the intermediate class $Y'$, in other words, given $\bar{Y}$, $Y'$ contains no more information for predicting $\bar{Y}$, then $Y$ is independent of $\bar{Y}$ conditioned on $Y'$, i.e.,

$$
T_{lj}^{\spadesuit}(Y = i) = P(\bar{Y} = j | Y' = l, Y = i) = P(\bar{Y} = j | Y' = l).
\tag{3}
$$

**Algorithm 1** dual-$T$ estimator
---
**Input:** Noisy training sample $S_{\text{tr}}$; Noisy validation sample $S_{\text{val}}$.

1: Obtain the learned noisy class posterior, i.e., $\hat{P}(\bar{\boldsymbol{Y}}|\boldsymbol{x})$, by exploiting training and validation sets;
2: Let $P(\boldsymbol{Y}'|\boldsymbol{x}) \triangleq \hat{P}(\bar{\boldsymbol{Y}}|\boldsymbol{x})$ and employ $T$ estimator to estimate $\hat{T}^{\clubsuit}$ according to Eq. (1);
3: Use Eq. (4) to estimate $\hat{T}^{\spadesuit}$;
4: $\hat{T} = \hat{T}^{\spadesuit}\hat{T}^{\clubsuit}$;

**Output:** The estimated transition matrix $\hat{T}$.

---

A sufficient condition for holding the above equalities is to let the intermediate class labels be identical to noisy labels. Note that it is hard to find an intermediate class whose labels are identical to noisy labels. The mismatch will be the main factor that contributes to the estimation error for $T^{\spadesuit}$. Note also that since we have labels for the noisy class and intermediate class, $P(\bar{Y} = j|Y' = l)$ in Eq. (3) is easy to estimate by just counting the discrete labels, and it will have a small estimation error which converges to zero exponentially fast [3].

Based on the above discussion, by factorizing the transition matrix $T$ into $T^{\spadesuit}$ and $T^{\clubsuit}$, we can change the problem of estimating the noisy class posterior into the problem of fitting the noisy labels. Note that the noisy class posterior is in the range of $[0, 1]$ while the noisy class labels are in the set $\{1, \dots, C\}$. Intuitively, learning the class labels are much easier than learning the class posteriors. In Section 4, our empirical experiments on synthetic and real-world datasets further justify this by showing a significant error gap between the estimation error of the $T$ estimator and dual-$T$ estimator.

**Implementation of the dual-$T$ estimator.** The dual-$T$ estimator is described in Algorithm 1. Specifically, the transition matrix $T^{\clubsuit}$ can be easily estimated by letting $P(Y' = i|\boldsymbol{x}) \triangleq \hat{P}(\bar{Y} = i|\boldsymbol{x})$ and then employing the $T$ estimator (see Section 2). By generating intermediate class labels, e.g., letting $\arg\max_{i \in \{1,\dots,C\}} P(Y' = i|\boldsymbol{x})$ be the label for the instance $\boldsymbol{x}$, the transition matrix $T^{\spadesuit}$ can be estimating via counting, i.e.,

$$\hat{T}^{\spadesuit}_{lj} = \hat{P}(\bar{Y} = j|Y' = l) = \frac{\sum_i \mathbb{1}_{\{(\arg\max_k P(Y'=k|\boldsymbol{x}_i)=l) \wedge \bar{y}_i=j\}}}{\sum_i \mathbb{1}_{\{\arg\max_k P(Y'=k|\boldsymbol{x}_i)=l\}}}, \tag{4}$$

where $\mathbb{1}_{\{A\}}$ is an indicator function which equals one when $A$ holds true and zero otherwise, $(\boldsymbol{x}_i, \bar{y}_i)$ are examples from the training sample $S_{\text{tr}}$, and $\wedge$ represents the AND operation.

Many statistically consistent algorithms [8, 26, 45, 37] consist of a two-step training procedure. The first step estimates the transition matrix and the second step builds statistically consistent algorithms, for example, via modifying loss functions. Our proposed dual-$T$ estimator can be seamlessly embedded into their frameworks. More details can be found in Section 4.

### 3.2 Theoretical Analysis

In this subsection, we will justify that the estimation error could be greatly reduced if we estimate $T^{\spadesuit}$ and $T^{\clubsuit}$ rather than estimating $T$ directly.

As we have discussed before, the estimation error of the $T$ estimator is caused by estimating the noisy class posterior; the estimation error of the dual-$T$ estimator comes from the estimation error of $T^{\spadesuit}$, i.e., fitting the noisy class labels and estimating $P(\bar{Y}|Y')$ by counting discrete labels. Note that to eliminate the dependence on the clean label for $T^{\spadesuit}$, we need to achieve $P(Y' = \bar{Y}|\boldsymbol{x}) = 1$. Let the estimation error for the noisy class posterior be $\Delta_1$, i.e., $\left|P(\bar{Y} = j|\boldsymbol{x}) - \hat{P}(\bar{Y} = j|\boldsymbol{x})\right| = \Delta_1$. Let the estimation error for $P(\bar{Y} = j|Y' = l)$ by counting discrete labels is $\Delta_2$, i.e., $|P(\bar{Y} = j|Y' = l) - \hat{P}(\bar{Y} = j|Y' = l)| = \Delta_2$. Let the estimation error for fitting the noisy class labels is $\Delta_3$, i.e., $P(Y' = \bar{Y}|\boldsymbol{x}) = 1 - \Delta_3$. We will show that under the following assumption, the estimation error of the dual-$T$ estimator is smaller than the estimation error the $T$ estimator.

**Assumption 1.** *For all $\boldsymbol{x} \in \bar{S}$, $\Delta_1 \geq \Delta_2 + \Delta_3$.*

Assumption 1 is easy to hold. Theoretically, the error $\Delta_2$ involves no predefined hypothesis space, and the probability that $\Delta_2$ is larger than any positive number will converge to zero exponentially fast [3]. Thus, $\Delta_2$ is usually much smaller than $\Delta_1$ and $\Delta_3$. We therefore focus on comparing $\Delta_1$

with $\Delta_3$ by ignoring $\Delta_2$. Intuitively, the error $\Delta_3$ is smaller than $\Delta_1$ because it is easy to obtain a small estimation error for fitting noisy class labels than that for estimating noisy class posteriors. We note that the noisy class posterior is in the continuous range of $[0, 1]$ while the noisy class labels are in the discrete set $\{1, \ldots, C\}$. For example, suppose we have an instance $(\boldsymbol{x}, \bar{y})$, then, as long as the empirical posterior probability $\hat{P}(\bar{Y} = \bar{y}|\boldsymbol{x})$ is greater than $1/C$, the noisy label will be accurately learned. However, the estimated error of the noisy class posterior probability can be up to $1 - 1/C$. We also empirically verify the relation among these errors in Appendix 2.

**Theorem 1.** *Under Assumption 1, the estimation error of the dual-$T$ estimator is smaller than the estimation error the $T$ estimator.*

## 4 Experiments

We compare the transition matrix estimator error produced by the proposed dual-$T$ estimator and the $T$ estimator on both synthetic and real-world datasets. We also compare the classification accuracy of state-of-the-art label-noise learning algorithms [19, 26, 13, 11, 37, 47, 23] obtained by using the $T$ estimator and the dual-$T$ estimator, respectively. The MNIST [16], Fashion-MINIST (or F-MINIST) [39], CIFAR10, CIFAR100 [15], and Clothing1M [40] are used in the experiments. Note that as there is no estimation error for $T^\clubsuit$, we do not need to do ablation study to show how the two new transition matrices contribute to the estimation error for transition matrix estimation.

### 4.1 Transition Matrix Estimation

We compare the estimation error between our estimator and the $T$ estimator on both synthetic and real-world datasets with different sample size and different noise types. The synthetic dataset is created by sampling from 2 different 10-dimensional Gaussian distributions. One of the distribution has unit variance and zero mean among all dimension. Another one has unit variance and mean of two among all dimensions. The real-world image datasets used to evaluate transition matrices estimation error are MNIST [16], F-MINIST [39], CIFAR10, and CIFAR100 [15].

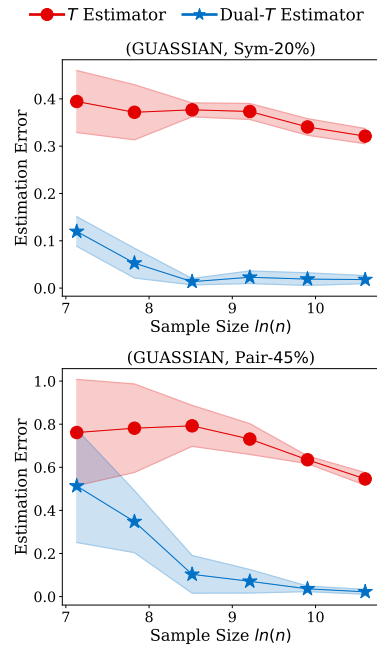

Figure 2: Estimation error of transition matrix on the synthetic dataset.

We conduct experiments on the commonly used noise types [11, 37]. Specifically, two representative structures of the transition matrix $T$ will be investigated: Symmetry flipping (Sym-$\epsilon$) [26]; (2) Pair flipping (Pair-$\epsilon$) [11]. To generate noisy datasets, we corrupt the training and validation set of each dataset according to the transition matrix $T$.

Neural network classifiers are used to estimate transition matrices. For fair comparison, the same network structure is used for both estimators. Specifically, on the synthetic dataset, a two-hidden-layer network is used, and the hidden unit size is 25; on the real-world datasets, we follow the network structures used by the state-of-the-art method [26], i.e., using a LeNet network with dropout rate $0.5$ for MNIST, a ResNet-18 network for F-MINIST and CIFAR10, a ResNet-34 network for CIFAR100, and a ResNet-50 pre-trained on ImageNet for Clothing1M. The network is trained for 100 epochs, and stochastic gradient descent (SGD) optimizer is used. The initial learning rate is $0.01$, and it is decayed by a factor 10 after $50$-th epoch. The estimation error is calculated by measuring the $\ell_1$-distance between the estimated transition matrix and the ground truth $T$. The average estimation error and the standard deviation over 5 repeated experiments for the both estimators is illustrated in Fig. 2 and 3.

It is worth mention that our methodology of anchor points estimation are different from the original paper of $T$ estimator [26]. Specifically, the original paper [26] includes a hyper-parameter to estimate anchor points for different datsets, e.g., selecting the points with $97\%$ and $100\%$ largest estimated noisy class posteriors to be anchor points on CIFAR-10 and CIFAR-100, respectively. However, clean data or prior information is needed to estimate the value of the hyper-parameter. By contrast, we

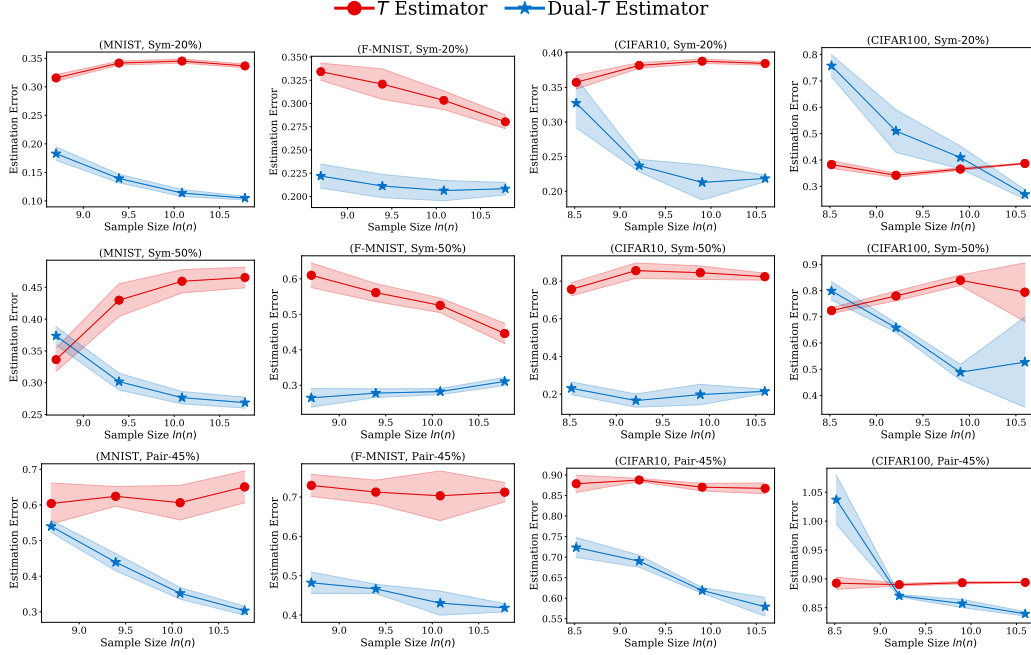

Figure 3: Transition matrix estimation error on MNIST, F-MNIST, CIFAR10, and CIFAR100. The error bar for standard deviation in each figure has been shaded. The lower the better.

assume that we only have noisy data, then it remains unclear to select the value of the hyper-parameter. In this case, from a theoretical point of view, we discard the hyper-parameter and select the points with the largest estimated intermediate class posteriors to be the anchor points for all datasets. This estimation method has been proven to be a consistent estimation of anchor points in binary case, and it also holds for multi-class classification when the noise rate is upper bounded by a constant [4]. Note that $T$ estimator [26] also selects the points with the largest estimated class posteriors to be anchor points on some datasets, which is the same as ours. Comparison on those datasets could directly justify the superiority of the proposed method. In addition, to estimate the anchor points, we need to estimate the noisy class posterior. However, deep networks are likely to overfit label noise [46, 1]. To prevent overfitting, we use $20\%$ training examples for validation, and the model with the best validation accuracy is selected for estimating anchor points. The anchor points are estimated on training sets.

Fig. 2 illustrates the estimation error of the $T$ estimator and the dual-$T$ estimator on the synthetic dataset. For two different noise types and sample sizes, the estimation error of the both estimation methods tend to decrease with the increasing of the training sample size. However, the estimation error of the dual-$T$ estimator is continuously smaller than that of the $T$ estimator. Moreover, the estimation error of the dual-$T$ estimator is less sensitive to different noise types compared to the $T$ estimator. Specifically, even the $T$ estimator is trained with all the training examples, its estimation error on Pair-$45\%$ noise is approximately doubled than that on Sym-20 noise, which is observed by looking at the right-hand side of the estimation error curves. In contrast, when training the dual $T$ estimator with all the training examples, its estimation error on the different noise types does not significantly different, which all less than $0.1$. Similar to the results on the synthetic dataset, the experiments on the real-world image datasets illustrated in Fig. 3 also shows that the estimation error of the dual-$T$ estimator is continuously smaller than that of the $T$ estimator except CIFAR100, which illustrates the effectiveness of the proposed dual-$T$ estimator. On CIFAR100, both estimators have a larger estimation error compared to the results on MNIST, F-MINIST, and CIFAR10. The dual-$T$ estimator outperforms the $T$ estimator with the large sample size. However, when the training sample size is small, the estimation error of the dual-$T$ estimator can be larger than that of the $T$ estimator. Because the numbers of images per class are too small to estimate the transition matrix $T^{\spadesuit}$, leading to a large estimation error.

## 4.2 Classification Accuracy Evaluation

| | MNIST | | | F-MNIST | | |
|---|---|---|---|---|---|---|
| | Sym-20% | Sym-50% | Pair-45% | Sym-20% | Sym-50% | Pair-45% |
| CE | $95.77 \pm 0.11$ | $93.99 \pm 0.21$ | $90.11 \pm 0.96$ | $89.70 \pm 0.14$ | $87.22 \pm 0.29$ | $73.94 \pm 1.44$ |
| Mixup | $91.14 \pm 0.28$ | $77.18 \pm 2.89$ | $80.14 \pm 1.74$ | $91.82 \pm 0.09$ | $89.83 \pm 0.11$ | $86.98 \pm 0.85$ |
| Decoupling | $98.34 \pm 0.12$ | $63.70 \pm 0.52$ | $56.66 \pm 0.25$ | $92.03 \pm 0.37$ | $86.96 \pm 0.86$ | $70.87 \pm 2.00$ |
| $T$ MentorNet | $91.51 \pm 0.31$ | $\mathbf{81.59} \pm 3.25$ | $62.10 \pm 4.11$ | $87.18 \pm 0.31$ | $\mathbf{79.32} \pm 2.08$ | $49.65 \pm 3.18$ |
| $DT$ MentorNet | $\mathbf{96.73} \pm 0.07$ | $78.99 \pm 0.4$ | $\mathbf{85.27} \pm 1.19$ | $\mathbf{92.93} \pm 0.07$ | $75.67 \pm 0.31$ | $\mathbf{81.84} \pm 1.34$ |
| $T$ Coteaching | $93.41 \pm 0.15$ | $\mathbf{84.13} \pm 2.77$ | $63.60 \pm 3.10$ | $88.10 \pm 0.29$ | $\mathbf{83.43} \pm 0.41$ | $58.18 \pm 7.00$ |
| $DT$ Coteaching | $\mathbf{97.52}^{*} \pm 0.07$ | $83.20 \pm 0.43$ | $\mathbf{86.78} \pm 0.76$ | $\mathbf{93.90}^{*} \pm 0.06$ | $77.45 \pm 0.59$ | $\mathbf{87.37} \pm 1.13$ |
| $T$ Forward | $96.85 \pm 0.07$ | $95.22 \pm 0.13$ | $94.92 \pm 0.89$ | $90.99 \pm 0.16$ | $88.58 \pm 0.30$ | $82.50 \pm 3.45$ |
| $DT$ Forward | $\mathbf{97.24} \pm 0.07$ | $\mathbf{95.89} \pm 0.14$ | $\mathbf{97.24}^{*} \pm 0.10$ | $\mathbf{91.37} \pm 0.09$ | $\mathbf{89.52} \pm 0.27$ | $\mathbf{91.91}^{*} \pm 0.24$ |
| $T$ Reweighting | $96.80 \pm 0.05$ | $95.25 \pm 0.23$ | $91.50 \pm 1.27$ | $90.94 \pm 0.29$ | $88.82 \pm 0.52$ | $80.94 \pm 3.38$ |
| $DT$ Reweighting | $\mathbf{97.34} \pm 0.04$ | $\mathbf{96.19} \pm 0.13$ | $\mathbf{96.62} \pm 0.21$ | $\mathbf{91.68} \pm 0.21$ | $\mathbf{90.17} \pm 0.12$ | $\mathbf{88.31} \pm 1.76$ |
| $T$ Revision | $96.79 \pm 0.04$ | $95.26 \pm 0.21$ | $91.83 \pm 1.08$ | $91.20 \pm 0.12$ | $88.77 \pm 0.36$ | $85.26 \pm 5.29$ |
| $DT$ Revision | $\mathbf{97.40} \pm 0.04$ | $\mathbf{96.21}^{*} \pm 0.13$ | $\mathbf{96.71} \pm 0.12$ | $\mathbf{91.78} \pm 0.16$ | $\mathbf{90.18}^{*} \pm 0.10$ | $\mathbf{90.70} \pm 0.37$ |
| | CIFAR10 | | | CIAR100 | | |
| | Sym-20% | Sym-50% | Pair-45% | Sym-20% | Sym-50% | Pair-45% |
| CE | $69.37 \pm 0.47$ | $55.92 \pm 0.44$ | $46.47 \pm 1.81$ | $33.16 \pm 0.56$ | $22.65 \pm 0.37$ | $21.62 \pm 0.58$ |
| Mixup | $80.33 \pm 0.59$ | $61.10 \pm 0.26$ | $58.37 \pm 2.66$ | $47.79 \pm 0.91$ | $30.17 \pm 0.74$ | $30.34 \pm 0.72$ |
| Decoupling | $81.63 \pm 0.34$ | $57.63 \pm 0.47$ | $52.30 \pm 0.16$ | $48.51 \pm 0.61$ | $26.01 \pm 0.40$ | $33.13 \pm 0.49$ |
| $T$ MentorNet | $79.00 \pm 0.20$ | $31.09 \pm 3.99$ | $26.19 \pm 2.24$ | $50.09 \pm 0.28$ | $36.66 \pm 9.13$ | $20.14 \pm 0.77$ |
| $DT$ MentorNet | $\mathbf{88.07} \pm 0.54$ | $\mathbf{69.34} \pm 0.61$ | $\mathbf{69.31} \pm 1.90$ | $\mathbf{59.7} \pm 0.41$ | $\mathbf{37.23} \pm 5.69$ | $\mathbf{30.88} \pm 0.58$ |
| $T$ Coteaching | $79.47 \pm 0.20$ | $39.71 \pm 3.52$ | $33.96 \pm 3.24$ | $50.87 \pm 0.77$ | $38.09 \pm 8.63$ | $24.58 \pm 0.70$ |
| $DT$ Coteaching | $\mathbf{90.37}^{*} \pm 0.12$ | $\mathbf{71.49} \pm 0.65$ | $\mathbf{76.51} \pm 4.97$ | $\mathbf{60.63}^{*} \pm 0.36$ | $\mathbf{38.21}^{*} \pm 5.91$ | $\mathbf{35.46}^{*} \pm 0.33$ |
| $T$ Forward | $75.36 \pm 0.39$ | $65.32 \pm 0.57$ | $54.70 \pm 3.07$ | $37.45 \pm 0.54$ | $27.91 \pm 1.48$ | $25.10 \pm 0.77$ |
| $DT$ Forward | $\mathbf{78.36} \pm 0.34$ | $\mathbf{69.94} \pm 0.66$ | $\mathbf{55.75} \pm 1.53$ | $\mathbf{41.76} \pm 0.97$ | $\mathbf{32.69} \pm 0.73$ | $\mathbf{26.08} \pm 0.93$ |
| $T$ Reweighting | $73.28 \pm 0.44$ | $64.20 \pm 0.38$ | $50.19 \pm 1.10$ | $38.07 \pm 0.34$ | $27.26 \pm 0.50$ | $\mathbf{25.86} \pm 0.55$ |
| $DT$ Reweighting | $\mathbf{79.09} \pm 0.21$ | $\mathbf{73.29} \pm 0.23$ | $\mathbf{52.65} \pm 2.25$ | $\mathbf{41.04} \pm 0.72$ | $\mathbf{34.56} \pm 1.39$ | $25.84 \pm 0.42$ |
| $T$ Revision | $75.71 \pm 0.93$ | $65.66 \pm 0.44$ | $75.14 \pm 2.43$ | $38.25 \pm 0.27$ | $27.70 \pm 0.64$ | $25.74 \pm 0.44$ |
| $DT$ Revision | $\mathbf{80.45} \pm 0.39$ | $\mathbf{73.76}^{*} \pm 0.22$ | $\mathbf{77.72}^{*} \pm 1.80$ | $\mathbf{42.11} \pm 0.76$ | $\mathbf{35.09} \pm 1.44$ | $\mathbf{26.10} \pm 0.43$ |

Table 1: Classification accuracy (percentage) on MNIST, F-MNIST, CIFAR10, and CIFAR100.

| CE | Mixup | Decoupling | $T(DT)$ MentorNet | $T(DT)$ Coteaching | $T(DT)$ Forward | $T(DT)$ Reweighting | $T(DT)$ Revision |
|---|---|---|---|---|---|---|---|
| 69.03 | 71.29 | 54.63 | 57.63 (**60.25**) | 60.37 (**64.54**) | 69.93 (**70.17**) | 70.38 (**70.86**) | 71.01 (**71.49**$^{*}$) |

Table 2: Classification accuracy (percentage) on Clothing1M.

We investigate how the estimation of the $T$ estimator and the dual-$T$ estimator will affect the classification accuracy in label-noise learning. The experiments are conducted on MNIST, F-MINIST, CIFAR10, CIFAR100, and Clothing1M. The classification accuracy are reported in Table 1 and Table 2. Eight popular baselines are selected for comparison, i.e., Coteaching [11], and MentorNet [13] which use diagonal entries of the transition matrix to help selecting reliable examples used for training; Forward [26], and Revision [37], which use the transition matrix to correct hypotheses; Reweighting [19], which uses the transition matrix to build risk-consistent algorithms. There are three baselines without requiring any knowledge of the transition matrix, i.e., CE, which trains a network on the noisy sample directly by using cross entropy loss; Decoupling [23], which trains two networks and updates the parameters only using the examples which have different prediction from two classifiers; Mixup [47] which reduces the memorization of corrupt labels by using linear interpolation to feature-target pairs. The estimation of the $T$ estimator and the dual-$T$ estimator are both applied to the baselines which rely on the transition matrix. The baselines using the estimation of $T$ estimator are called $T$ Coteaching, $T$ MentorNet, $T$ Forward, $T$ Revision, and $T$ Reweighting. The baselines using estimation of dual-$T$ estimator are called $DT$ Coteaching, $DT$ MentorNet, $DT$ Forward, $DT$ Revision, and $DT$ Reweighting.

The settings of our experiments may be different from the original paper, thus the reported accuracy can be different. For instance, in the original paper of Coteaching [11], the noise rate is given, and all data are used for training. In contrast, we assume the noise rate is unknown and needed to be estimated. We only use $80\%$ data for training, since $20\%$ data are leaved out as the validation set for transition matrix estimation. In the original paper of $T$ revision [37], the experiments on Clothing1M use clean data for validation. In contrast, we only use noisy data for validation.

In Table 1 and Table 2, we bold the better classification accuracy produced by the baseline methods integrated with the $T$ estimator or the dual-$T$ estimator. The best classification accuracy among all the methods in each column is highlighted with $*$. The tables show the classification accuracy of all the methods by using our estimation is better than using that of the $T$ estimator for most of the experiments. It is because that the dual-$T$ estimator leads to a smaller estimation error than the $T$ estimator when training with large sample size, which can be observed at the right-hand side of the estimation error curves in Fig. 3. The baselines with the most significant improvement by using our estimation are Coteaching and MentorNet. $DT$ Coteaching outperforms all the other methods under Sym-20% noise. On Clothing1M dataset, $DT$ revision has the best classification accuracy. The experiments on the real-world datasets not only show the effectiveness of the dual-$T$ estimator for improving the classification accuracy of the current noisy learning algorithms, but also reflect the importance of the transition matrix estimation in label-noise learning.

## 5    Conclusion

The transition matrix $T$ plays an important role in label-noise learning. In this paper, to avoid the large estimation error of the noisy class posterior leading to the poorly estimated transition matrix, we have proposed a new transition matrix estimator named dual-$T$ estimator. The new estimator estimates the transition matrix by exploiting the divide-and-conquer paradigm, i.e., factorizes the original transition matrix into the product of two easy-to-estimate transition matrices by introducing an intermediate class state. Both theoretical analysis and experiments on both synthetic and real-world label noise data show that our estimator reduces the estimation error of the transition matrix, which leads to a better classification accuracy for the current label-noise learning algorithms.

## Broader Impact

In the era of big data, it is expensive to expert-labeling each instance on a large scale. To reduce the annotation cost of the large datasets, non-expert labelers or automated annotation methods are widely used to annotate the datasets especially for startups and non-profit organizations which need to be thrifty. However, these cheap annotation methods are likely to introduce label-noise to the datasets, which put threats to the traditional supervised learning algorithm, and label-noise learning algorithms, therefore, become more and more popular.

The transition matrix which contains knowledge of the noise rates is essential to building the classifier-consistent label-noise learning algorithms. The classification accuracy of the state-of-the-art label-noise learning applications are usually positively correlated to the estimation accuracy of the transition matrix. The proposed method is to estimate the transition matrix of noisy datasets. We have shown that our method usually leads to a better estimation compared to the current estimator and improves the classification accuracy of current label-noise learning applications. Therefore, outcomes of this research including improving the accuracy, robustness, accountability of the current label-noise learning applications, which should benefit science, society, and economy internationally. Potentially, our method may have a negative impact on the job of annotators.

Because the classification accuracy of the label-noise learning applications is usually positively correlated to the estimation accuracy of the transition matrix. Therefore, the failure of our method may lead to a degenerating of the performance of the label-noise learning applications.

The proposed method does not leverage biases in the data.

## Acknowledgments

TL was supported by Australian Research Council Project DE-190101473. BH was supported by the RGC Early Career Scheme No. 22200720, NSFC Young Scientists Fund No. 62006202, HKBU

Tier-1 Start-up Grant and HKBU CSD Start-up Grant. GN and MS were supported by JST AIP Acceleration Research Grant Number JPMJCR20U3, Japan. The authors thank the reviewers and the meta-reviewer for their helpful and constructive comments on this work.

## Footnotes

[1]Because the noisy labels come from taking the argmax of the noisy posteriors, where the argmax compresses information. Thus, the noisy labels contain much less information than the noisy posterior, and predicting the noisy labels require much less information than estimating the noisy posteriors.

[2]If the anchor points are to learn, the estimation error remains unchanged for the $T$ estimator and dual-$T$ estimator by employing $\boldsymbol{x}^i = \arg\max_{\boldsymbol{x}} \hat{P}(\bar{Y} = i | \boldsymbol{x})$.

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
