[Supplementary Material]

# Supplementary to "Dual T: Reducing Estimation Error for Transition Matrix in Label-noise Learning"

## 1 Proof of Theorem 1

*Proof.* According to Eq. (1) in the main paper, the estimation error for the $T$ estimator is

$$\epsilon_T = \sum_{i,j} \left| T_{ij} - \hat{T}_{ij} \right| = \sum_{i,j} \left| P(\bar{Y} = j | X = \boldsymbol{x}^i) - \hat{P}(\bar{Y} = j | X = \boldsymbol{x}^i) \right|. \tag{1}$$

As we have assumed, for all instance $\boldsymbol{x} \in \mathcal{X}$, for all $j \in \{1, \ldots, C\}$,

$$\left| P(\bar{Y} = j | X = \boldsymbol{x}) - \hat{P}(\bar{Y} = j | X = \boldsymbol{x}) \right| = \Delta_1. \tag{2}$$

Then, we have

$$\epsilon_T = C^2 \Delta_1. \tag{3}$$

The estimation error for the $i, j$-the entry of the dual-$T$ estimator is

$$\begin{aligned}
&\left| \sum_l P(\bar{Y} = j | Y' = l, Y = i) P(Y' = l | Y = i) \right. \\
&\left. - \sum_l \hat{P}(\bar{Y} = j | Y' = l) P(Y' = l | Y = i) \right| \\
&= \sum_l \left| P(\bar{Y} = j | Y' = l, Y = i) - \hat{P}(\bar{Y} = j | Y' = l) \right| P(Y' = l | Y = i),
\end{aligned} \tag{4}$$

where the first equation holds because there is no estimation error for the transition matrix denoting the transition from the clean class to the intermediate class (as we have discussed in Section 3.1). The estimation error for the dual-$T$ estimator comes from the estimation error for fitting the noisy class labels (to eliminate the dependence on the clean label) and the estimation error for $P(\bar{Y} = j | Y' = l)$ by counting discrete labels.

We have assumed that the estimation error for $P(\bar{Y} = j | Y' = l)$ is $\Delta_2$, i.e., $|P(\bar{Y} = j | Y' = l) - \hat{P}(\bar{Y} = j | Y' = l)| = \Delta_2$ and that the estimation error for fitting the noisy class labels is $\Delta_3$, i.e., $\forall \boldsymbol{x} \in \mathcal{X}, P(Y' = \bar{Y} | \boldsymbol{x}) = 1 - \Delta_3$. Note that, to eliminate the dependence on the clean label for $T^{\spadesuit}$, we need to achieve $P(Y' = \bar{Y} | \boldsymbol{x}) = 1$ for all $\boldsymbol{x} \in \mathcal{X}$. The error $\Delta_3$ will be introduced if there is an error for fitting the noisy class labels. We have that $P(Y' \neq \bar{Y} | \boldsymbol{x}) = \Delta_3$.

We have

$$\left| P(\bar{Y}=j|Y'=l,Y=i) - \hat{P}(\bar{Y}=j|Y'=l) \right|$$

$$= \left| P(\bar{Y}=j|Y'=l,Y=i,\boldsymbol{x}) - \hat{P}(\bar{Y}=j|Y'=l,\boldsymbol{x}) \right|$$

$$= \left| P(\bar{Y}=j|Y'=l,Y=i,\boldsymbol{x})P(Y'=\bar{Y}|\boldsymbol{x}) + P(\bar{Y}=j|Y'=l,Y=i,\boldsymbol{x})P(Y'\neq\bar{Y}|\boldsymbol{x}) \right.$$
$$\left. - \hat{P}(\bar{Y}=j|Y'=l,\boldsymbol{x})P(Y'=\bar{Y}|\boldsymbol{x}) - \hat{P}(\bar{Y}=j|Y'=l,\boldsymbol{x})P(Y'\neq\bar{Y}|\boldsymbol{x}) \right|$$

$$= \left| P(\bar{Y}=j|Y'=l,\boldsymbol{x})P(Y'=\bar{Y}|\boldsymbol{x}) + P(\bar{Y}=j|Y'=l,Y=i,\boldsymbol{x})P(Y'\neq\bar{Y}|\boldsymbol{x}) \right.$$
$$\left. - \hat{P}(\bar{Y}=j|Y'=l,\boldsymbol{x})P(Y'=\bar{Y}|\boldsymbol{x}) - \hat{P}(\bar{Y}=j|Y'=l,\boldsymbol{x})P(Y'\neq\bar{Y}|\boldsymbol{x}) \right| \qquad (5)$$

$$\leq \left| P(\bar{Y}=j|Y'=l) - \hat{P}(\bar{Y}=j|Y'=l) \right| P(Y'=\bar{Y}|\boldsymbol{x})$$
$$+ \left| P(\bar{Y}=j|Y'=l,Y=i) - \hat{P}(\bar{Y}=j|Y'=l) \right| P(Y'\neq\bar{Y}|\boldsymbol{x})$$
$$\leq \Delta_2(1-\Delta_3) + \Delta_3 < \Delta_2 + \Delta_3,$$

where the second equation holds because the transition matrices are independent of instances. Hence, the estimation error of $T^\spadesuit$ is

$$\epsilon_{DT} = \sum_{i,j,l} \left| P(\bar{Y}=j|Y'=l,Y=i) - \hat{P}(\bar{Y}=j|Y'=l) \right| P(Y'=l|Y=i)$$

$$< \sum_{i,j} \sum_{l} (\Delta_2+\Delta_3)P(Y'=l|Y=i)$$

$$= \sum_{i,j} (\Delta_2+\Delta_3) = C^2(\Delta_2+\Delta_3). \qquad (6)$$

Therefore, under Assumption 1 in the main paper, the estimation error $\epsilon_{DT}$ of the dual-$T$ estimator is smaller than the estimation error $\epsilon_T$ the $T$ estimator. □

## 2 Empirical Validation of Assumption 1

We empirically verify the relations among the three different errors in Assumption 1 on different type of classifiers. Note that $\Delta_1$ is the estimation error for the noisy class posterior, i.e., $\Delta_1 = \left| P(\bar{Y}=j|\boldsymbol{x}) - \hat{P}(\bar{Y}=j|\boldsymbol{x}) \right|$; $\Delta_2$ is the estimation error for counting discrete labels, i.e., $|P(\bar{Y}=j|Y'=l) - \hat{P}(\bar{Y}=j|Y'=l)| = \Delta_2$; $\Delta_3$ is the estimation error for fitting the noisy class labels, i.e., $P(Y'=\bar{Y}|\boldsymbol{x}) = 1 - \Delta_3$.

The experiments are conducted on the synthetic dataset, and setting is same as those of the synthetic experiments in Section 4. The three errors are calculated on the training set, since both the $T$ estimator and the dual-$T$ estimator estimates the transition matrix on the training set. We employ a 16-layer neural network with short cut connection [2] for feature extraction. Then the extracted features are passed to a linear classifier by employing cross entropy loss and a logistic regression classifier by employing logistics loss, respectively.

Figure 1: The relations among $\Delta_1$, $\Delta_2$ and $\Delta_3$ by employing logistic loss function.

Figure 2: The relations among $\Delta_1$, $\Delta_2$ and $\Delta_3$ by employing cross entropy loss.

Both Figure 1 and Figure 2 show that the error $\Delta_2$ is very small and can be ignored. $\Delta_3$ is continuously smaller than $\Delta_1$ when the sample size is small. The recent work [1] shows that the sample complex of the network is linear in the number of parameters, which means that, usually, we may not have enough training examples to learn the noisy class posterior well (e.g., CIFAR10, CIFAR100, and Fashion-MNIST), and Assumption 1 can be easily satisfied. It is also worth to mention that, even Assumption 1 does not hold, the estimation error of the dual-$T$ estimator may also be smaller than the $T$ estimator. Specifically, the error $\epsilon_{DT}$ of the proposed estimator is upper bounded by $C^2(\Delta_2 + \Delta_3)$. Generally, the increasing of the upper bound $C^2(\Delta_2 + \Delta_3)$ does not imply the increasing of the error $\epsilon_{DT}$. Figure 1 and Figure 2 also show that different types of the classifiers have different capabilities to fit the noisy labels. Specifically, the linear classifier has smaller error $\Delta_3$ on pair flipping $45\%$ compared to that on symmetry flipping $20\%$, by contrast, the logistic regression classifier has larger error $\Delta_3$ on pair flipping $45\%$ compared to that on symmetry flipping $20\%$.