[Reviews · NeurIPS 2020]

Review 1

Summary and Contributions: The authors introduce an algorithm Dual T, a divide and conquer approach for estimating transition matrices which is in essential in building classifier consistent algorithm for label noise learning. Dual T works by decomposing the problem into two sub-problems of estimating two transition matrices in order to reduce estimation error.

Strengths: The work is well motivated and builds off other established methods. The empirical evaluation is sound and seem to provide evidence for the performance of the method. The authors also compare their method to different baselines.

Weaknesses: The main weakness of the paper is based on the assumption in the methodology. The authors claim that that the estimation error for T^{club} transition matrix is zero if anchor points are given. However, in practice this is not easily obtained as the authors stated. The authors did not explain how their method will perform when anchor points are not provided or estimated.

Correctness: The authors do provide theoretical justifications under some assumptions on why errors from Dual T will be lower than directly estimating the transition matrix. The empirical methodology is good however it will be better if the authors did an ablation study for the errors of the two matrices.

Clarity: The paper is mostly well written and easy to follow however section 3.1 in the methodology can be modified to make it easier to understand.

Relation to Prior Work: Yes, the authors describe how this work is differs from prior works however a core part of the methodology is based on previous work.

Reproducibility: Yes

Additional Feedback: Based on my comments above, I have a a few questions 1. Will the transition matrix for for T^{club} be zero if the anchor points are estimated, if so why? 2. We assume that the intermediate class labels are provided, however this is given as the estimated noise posterior. Won't this have an estimation error that will affect the error of both matrices? 3. On the synthetic dataset, we see that T estimator performs better than Dual T on small datasets however we see in Table 1 that on large real datasets, T estimator can outperform Dual T. Is there an explanation on why this happens? ========================================= I acknowledge that I read the rebuttal and thank the authors for providing explanations to the questions and concerns I had.


Review 2

Summary and Contributions: This paper studies how to effectively estimate the transition matrix in label-noise learning. This paper first shows that the estimation error for noisy class posterior could be large due to the randomness of label noise, so that the transition matrix estimated by the original anchor point based method would be inaccurate. To address this problem, this paper introduces an intermediate class and estimates two decomposed matrices, which are easier to be estimated. Both theoretical analyses and empirical results demonstrate the effectiveness of the proposed dual T method. It is a very interesting paper that not only reveals the problem of the original anchor point based estimation method, but also proposes an effective method to alleviate this issue. Theoretical analyses and empirical results support this method. After rebuttal: I have read the comments and rebuttal. The authors have well explained Eq. 3 and the issue about validation set, and also provide new experimental results. My final recommendation is still accept because the method proposed is well-motivated, novel, and interesting, which would interest many readers.

Strengths: 1. It reveals the problem of the original anchor point based estimation method. 2. It proposes an effective method to alleviate this issue. 3. Theoretical analyses and empirical results support this method.

Weaknesses: 1. There are several typos in this paper. For example, in line 123, "we can assess to the the noisy class...". The title of Section 4.2, "Classification accuracy Evaluation" where "accuracy" should be "Accuracy". 2. I understand that Assumption 1 is easy to hold, while it would be better to prove Theorem 1 without this assumption. I suggest the authors could discuss how we might improve that from some aspects for furture work.

Correctness: Yes

Clarity: Yes

Relation to Prior Work: Yes

Reproducibility: Yes

Additional Feedback: Please check the weaknesses.


Review 3

Summary and Contributions: The paper deals with the topic of learning with noisy labels in the context of statistically consistent classifiers. The authors propose a new approach (called Dual T-estimator) for estimating the transition matrix that can be used to infer the clean class posterior from the noisy class posterior. They focus on the setting where the transition matrix is assumed to be class-dependent and instance-independent. In the Dual T-estimator approach, instead of directly estimating the transition matrix, two "intermediate" matrices are estimated and the product of those two matrices is used as an estimate of the transition matrix. The authors provide a proof showing that the estimation error of their approach is lower than that obtained when directly estimating the transition matrix.

Strengths: The transition matrix can be estimated by having a model trained on noisy class labels and by having a set of (given or estimated) anchor points. Depending on the amount and type of noise in the labels and the number of training examples, a model trained on noisy class labels will overfit the noise. This can lead to a high estimation error of the transition matrix (whose elements are computed by estimating the posterior of the noisy labels of anchor points). The authors propose an approach to revise / re-adjust the transition matrix. The idea seems sound. Note that the concept of revising the transition matrix is not new. But the described approach for doing so seems to be new.

Weaknesses: 1) The justification and explanation of equation 3 (which is a central point of the paper) is not clear. Here's how I interpret the approach proposed by the authors based on Algorithm 1. The elements of the Dual T-estimator transition matrix are computed as follows: \hat{T}_{ij} = \sum_l \hat{P}(\bar{Y}=j | Y'=l) \hat{P}(Y'=l | Y_i) The second element in the sum is obtained using equation 1, which is the same equation used to compute the T-estimator of the transition matrix. The first element in the sum is estimated using equation 4 which determines the number of examples belonging to the noisy class j which were incorrectly labeled as belonging to the noisy class l divided by the number of examples labeled as belonging to the noisy class l. This ratio is basically used to revise / correct the T-estimator of the transition matrix (and hence mitigate the effect of overfitting the noise in the training set). Whether my interpretation is the one the authors had in mind or not, I think the justification or explanation of equation 3 should be clarified. 2) The empirical evaluation section is lacking a section that discusses the results and provides insights.

Correctness: Yes.

Clarity: Clarity of the paper needs to be improved: * There are some repetitions of text that don't necessarily bring added value. For example, lines 152-156 are repetitions of earlier lines in the paper. * There are several grammatical mistakes. Following are some examples: - Line 19: add an "s" to "memorize" and "fit". - Line 43: replace "accurate" with "accurately". - Caption of figure 1: replace "absolute value" with "absolute difference". - Line 62: replace "In a high level" with "At a high level". - Line 108: replace "to product" with "into the product". - Line 116: replace "is focused by" with "is the focus of the". - Line 119: replace "accurate" with "accurately". - Line 123: "can assess" should probably be replaced with "have access". - Line 132: remove the comma after "applications". - Line 148: replace "represent" with "represents". ... * The value of lines 105-110 is unclear. * Lines 173 and 174 seem to have the wrong sequence of symbols. I think the text should be: "in other words, given Y', Y contains no more information for predicting \bar{Y}".

Relation to Prior Work: There isn't a related work section in the paper. And there isn't a discussion of how the presented work differs from previous contributions. There are other methods that try to correct the estimated transition matrix. The authors don't discuss how their method differs and don't describe the pros/cons of various correction approaches.

Reproducibility: Yes

Additional Feedback: I read the authors' rebuttal. They clarified a misunderstanding I had about the data used in equation 4. They also ran additional experiments with a smaller validation set (10% instead of 20%) and showed that the DualT approach continues to be useful. And they clarified the reasons for which they chose an experimental protocol that is different from the ones used in the papers describing other approaches. Based on the rebuttal, I changed my decision from reject to accept.


Review 4

Summary and Contributions: The paper proposes dual T-estimator, a new transition matrix estimator, for deep learning with noisy labels. It factorizes the original transition matrix into the product of two easy-to-estimate transition matrices. Theoretical analysis and experiments show that it can reduce the estimation error and provide better classification accuracy.

Strengths: The paper proposes a novel estimator for transition matrix estimation, which plays an important role in designing statistically consistent algorithms in learning with noisy labels settings. It is theoretically proved that the proposed estimator has smaller estimation error than the traditional estimator under an easy-to-hold assumption. The proposed estimator can be easily embedded to those statistically consistent algorithms, which usually have a step to estimate the transition matrix. Experiments show that the proposed estimator can reduce estimation error comparing to the traditional estimator and provide better classification accuracy when embedded to those statistically consistent algorithms.

Weaknesses: The estimation error of the proposed estimator mainly comes from (1) fitting the noisy labels and (2) estimating P(\bar{Y} | Y'). Since (2) needs counting discrete labels, it may require large sample size for each class. Experiments also show that when the number of class is big (e.g. CIFAR-100) and the sample size is small, the estimation error of the proposed estimator can be bigger than that of the traditional estimator. There is little discussion about the impact of (1) in the paper. e.g. will different types of the classifiers, which have different capabilities to fit the noisy labels, influence the performance of the estimator and how large the influence can be?

Correctness: Yes.

Clarity: Yes.

Relation to Prior Work: The paper only compares the proposed estimator with the traditional T-estimator theoretically and empirically. It is not clear whether there are other transition matrix estimation algorithms and whether the proposed estimator is better than them in some aspects.

Reproducibility: Yes

Additional Feedback: Line 124 and Equation (1), P(\bar{Y}=j | Y=i, x) - Should it be x^i instead of x? Line 173, in other words, ... - Should it be "given Y', Y contains no more information for predicting \bar{Y}"? After rebuttal: I have read the author feedback. It well addresses my question about fitting the noisy labels. Additional empirical study will be interesting. After the rebuttal, I still think the paper is a good submission and I would like to accept it.

[Author Response · NeurIPS 2020]

We sincerely thank all reviewers for their constructive comments and positive support. In particular, we respectively ask
R3 to reconsider his/her evaluation since we can address all the comments, as detailed below.

**Q1 (R1):** Will the estimation error for $T^{\clubsuit}$ be zero if the anchor points are estimated, if so why? And ablation study.
**A1:** There will be an estimation error for $T^{\clubsuit}$ if there is an estimation error for the anchor points. The same estimation
error for the anchor points will also go for the $T$-estimator. For simplicity, we study the setting where anchor points are
given. In this case, the error only comes from estimating $T^{\spadesuit}$. Thus we do not need to do ablation study for the errors
of the two matrices $T^{\clubsuit}$ and $T^{\spadesuit}$. Note that even given anchor points, there is an estimation error for the $T$-estimator
because of estimating the noisy class posterior while there is no estimation error for $T^{\clubsuit}$ because the intermediate class
posterior is designed to be given. Considering how the estimation error for anchor points affects the two estimators is
also very interesting. We leave it as a future work.

**Q2 (R1):** Will intermediate class labels obtained ... have an estimation error that will affect the error of both matrices?
**A2:** The intermediate class labels are defined by the ones that maximize the intermediate class posteriors. Since we
have given the true intermediate class posterior distribution, i.e., $P(Y'|\boldsymbol{x}) = \hat{P}(\bar{Y}|\boldsymbol{x})$, there will be no estimation error
for the intermediate class labels.

**Q3 (R1):** On the synthetic dataset, dual T-estimator performs better ... Is there an explanation on why this happens?
**A3:** On the real-world datasets, the estimation error of the dual $T$-estimator can be larger than that of the $T$-estimator
only with small training sample sizes. It is because the number of images per class is too small to estimate the transition
matrix $\hat{T}^{\spadesuit}$ which can be sparse and lead to a large estimation error. We will discuss this in the final version.

**Q4 (R3):** The justification and explanation of Eq. 3 is not clear. $\hat{P}(\bar{Y}|Y')$ (which is computed using the validation set).
**A4:** We will add more discussions about Eq.3. It discusses how to make $T^{\spadesuit}$ to be independent of $Y$ because $Y$ is unavail-
able. Specifically, we have explained a sufficient condition for letting Eq. 3 hold, i.e., let the intermediate class labels be
identical to noisy labels. We have also discussed that the condition may be hard to be satisfied, therefore, an estimation
error for fitting the noisy labels to intermediate class labels $\Delta_3$ is introduced to our estimation. In Appendix 2, we have
also empirically validated the estimation error. In line 193, we have stated that $\hat{P}(\bar{Y}|Y')$ is NOT computed on the valida-
tion set. It is computed on the training set. The validation set functions as the same role for both the proposed method and
the baselines for model selection. Your concerns on the unfair training/validation split may not stand. However, we agree
that to see how the size of validation set influences the performance is interesting. We have redone experiments according
to your suggestions to reduce the size of validation set, i.e., using 10% of the training examples as a validation set for the
both estimators with five repeated trials (random seeds are from 1 to 5). The estimation errors for transition matrices are
illustrated in the following table, which show that the dual $T$-estimator still significantly outperforms the $T$-estimator.

|  | MNIST | | | F-MNIST | | | CIFAR-10 | | | CIFAR-100 | | |
|---|---|---|---|---|---|---|---|---|---|---|---|---|
|  | Sym-20% | Sym-50% | Pair-45% | Sym-20% | Sym-50% | Pair-45% | Sym-20% | Sym-50% | Pair-45% | Sym-20% | Sym-50% | Pair-45% |
| $T$ | 0.333±0.002 | 0.459±0.001 | 0.644±0.069 | 0.261±0.012 | 0.428±0.019 | 0.687±0.097 | 0.383±0.004 | 0.772±0.077 | 0.861±0.028 | 0.389±0.001 | 0.760±0.117 | 0.894±0.002 |
| Dual $T$ | **0.106**±0.003 | **0.271**±0.007 | **0.298**±0.021 | **0.220**±0.014 | **0.315**±0.012 | **0.414**±0.020 | **0.215**±0.011 | **0.267**±0.075 | **0.578**±0.026 | **0.255**±0.020 | **0.516**±0.187 | **0.835**±0.014 |

**Q5 (R3):** Why use T-Revision models of poorer classification accuracy in the Dual T-estimator paper?
**A5:** This could be a common concern if we focus on improving the classification accuracy. However, our aim is to
reduce the estimation error for the transition matrix. The experiments are set to FAIRLY verify the effectiveness of the
Dual $T$-estimator. As mentioned by the reviewer, the settings of our experiments are different from the original paper,
thus the reported accuracy is different. Specifically, to boost the classification performance, different tricks has been
employed in the baselines, e.g., transition information is given (e.g., Co-Teaching); choosing different hyper-parameters
to estimate anchor points (e.g., using the 97% and 100% largest estimated noisy class posteriors for *CIFAR-10* and
*CIFAR-100*, respectively, in Forward); training networks with different epochs to estimate noisy class posteriors (e.g.,
40 epochs in Forward while 20 epochs in T-revision on *mnist*); with or without using the validation set to estimate noisy
class posteriors (e.g., T-revision uses a validation set while Forward does not). To fairly show the superiority of Dual
$T$, all the baselines in our paper use the same tricks, e.g., using a noisy validation set to choose model for estimating
the noisy class posterior and using the largest noisy class posteriors for estimating anchor points. So the classification
performance would be worse than that in the original paper. However, the effectiveness of the Dual $T$-estimator has
been successfully illustrated. We will add the discussions in the final version.

**Q6 (R4):** There is little discussion about the impact of (1) in the paper. e.g. will different types of the classifiers, which
have different capabilities to fit the noisy labels, influence the performance of the estimator and how large?
**A6:** Thanks for the insightful question. We will add more discussions. For example, the estimation error for $T$-estimator
is from estimating the noisy class posterior in (1); the estimator error for Dual $T$-estimator (or $T^{\spadesuit}$ because $T^{\clubsuit}$ is
noise free) is also related to (1) because to push the intermediate class $Y'$ to be close to the noisy class $\bar{Y}$. Studying
how different classifiers influence the performance of the estimator via impacting (1) is interesting. In our paper, we
assume that $\Delta_1$, which is the estimation error for the noisy class posterior, is bigger than $\Delta_3$, which is the estimation
error for pushing the intermediate class $Y'$ to be equal to the noisy class $\bar{Y}$. Note that if $\Delta_1 \geq \Delta_2 + \Delta_3$, the dual
$T$-estimator will outperform $T$-estimator, where $\Delta_2 = |P(\bar{Y}|Y') - \hat{P}(\bar{Y}|Y')|$ is the error introduced by counting the
noisy and intermediate class labels. We discussed when this will hold and empirically validated this assumption in the
supplementary material. We agree with the reviewer that it is interesting to study when the assumption is invalid and
how different types of classifiers influence it. We would add some empirical study in the supplementary material.

[Meta-Review · NeurIPS 2020]

This main contribution is a study to estimate the transition matrix in label-noise learning. The paper proposes interesting methods, with sound theory and empirical study. The authors are strongly encouraged to, and we trust that you will, significantly improve the clarity of the discussion in the few places mentioned by the reviewers.